# Predicting the risk of cancer in adults using supervised machine learning: a scoping review

Asma Abdullah Alfayez [1,2] Holger Kunz [1] Alvina Grace Lai [1]

[1]Institute of Health Informatics, University College London, London, UK
[2]King Abdullah International Medical Research Center, Riyadh, Saudi Arabia

**Correspondence to**
Ms Asma Abdullah Alfayez;
asma.alfayez.17@ucl.ac.uk and
Dr Alvina Grace Lai;
alvina.lai@ucl.ac.uk

## ABSTRACT

**Objectives** The purpose of this scoping review is to: (1) identify existing supervised machine learning (ML) approaches on the prediction of cancer in asymptomatic adults; (2) to compare the performance of ML models with each other and (3) to identify potential gaps in research.

**Design** Scoping review using the population, concept and context approach.

**Search strategy** PubMed search engine was used from inception to 10 November 2020 to identify literature meeting following inclusion criteria: (1) a general adult (≥18 years) population, either sex, asymptomatic (population); (2) any study using ML techniques to derive predictive models for future cancer risk using clinical and/or demographic and/or basic laboratory data (concept) and (3) original research articles conducted in all settings in any region of the world (context).

**Results** The search returned 627 unique articles, of which 580 articles were excluded because they did not meet the inclusion criteria, were duplicates or were related to benign neoplasm. Full-text reviews were conducted for 47 articles and a final set of 10 articles were included in this scoping review. These 10 very heterogeneous studies used ML to predict future cancer risk in asymptomatic individuals. All studies reported area under the receiver operating characteristics curve (AUC) values as metrics of model performance, but no study reported measures of model calibration.

**Conclusions** Research gaps that must be addressed in order to deliver validated ML-based models to assist clinical decision-making include: (1) establishing model generalisability through validation in independent cohorts, including those from low-income and middle-income countries; (2) establishing models for all cancer types; (3) thorough comparisons of ML models with best available clinical tools to ensure transparency of their potential clinical utility; (4) reporting of model calibration performance and (5) comparisons of different methods on the same cohort to reveal important information about model generalisability and performance.

## Strengths and limitations of this study

► This study used a recognised scoping review approach (population, concept and context) to explore the machine learning techniques used to derive predictive models for future cancer risk.

► Identified studies were not subjected to comprehensive qualitative assessments.

► Only 10 studies were identified, making it difficult to draw firm conclusions about their relative performance.

► Area under the curve values (AUC) alone do not allow for meaningful comparisons of models as they have been trained and evaluated on different datasets under different circumstances and conditions.

► This scoping review is limited to papers published in English until 2020 and only the PubMed search engine was used.

## INTRODUCTION

Cancer remains a leading cause of morbidity and mortality, with an estimated 1.8 million new cases and 0.6 million deaths in the USA in 2019 and approximately 367 000 new cases and 165 000 cancer deaths in the UK each year between 2015 and 2017.[1 2] Annual death rates only modestly decreased (1.4% and 1.8% in women and men, respectively) between 2012 and 2016, despite significant research.[1] Cancer cases also continue to increase, not least due to increased life expectancy, which increases the risk of developing cancer.[3]

Early cancer diagnosis is associated with significantly higher survival rate and lower mortality and associated costs. Early-stage cancers require less complex treatment regimens and reduced hospital utilisation, resulting in reduced healthcare costs, whereas late-stage cancers require complex multimodal management, several rounds of extremely expensive drugs over significant periods of time, and the treatment of recurrences, equating to a staggering economic burden. Therefore, the importance of early diagnosis cannot be overestimated.[4–6] Treating cancer early has significant cost-saving benefits. In the USA, during the first 24 months after diagnosis, there is an increase in cancer treatment costs with stage: US$72 000 for stage 0, US$97 000 for stage I/II, US$159 000 for stage III and US$182 000 for stage IV.[7] An estimate of the cost savings from early cancer diagnosis is US$26 billion

BMJ

per annum in the USA alone.[8] Similarly, in the UK, early diagnosis of colorectal, ovarian, and lung cancer in England alone could provide savings of over £44 million and benefit nearly 11 000 patients.[9]

Survival rates significantly improve if cancer is diagnosed at stage I or II compared with later stages (stage III and IV),[10 11] as once the cancer has metastasised, it becomes difficult to treat with radiotherapy or surgery, leading to treatment failure and death. For example, 5-year survival rates for women diagnosed with localised breast or ovarian cancer are 99% and 92% compared with 27% and 29% for metastatic disease, respectively.[1] A report by Cancer Research UK indicated that, in the UK, the 10-year survival proportions of patients with eight cancers (combined) were around 80% for stage I and stage II detection (breast, bladder, ovarian, colorectal, uterine, testicular, cervical, and malignant melanoma) but only 26% for cancers detected at later stages, notably lung cancer (stage III and IV).[12]

### Current approaches to diagnose incident cancer

One approach to the early detection of cancer is population-wide screening, which aims to find asymptomatic individuals so that they can be promptly referred for treatment. Examples include mammography for breast cancer, cervical screening for cervical cancer and faecal occult blood testing or sigmoidoscopy for colorectal cancer.[13] There are three examples of national screening programmes in UK (bowel, breast and cervical cancer screening programs[14] and two in the USA: the Colorectal Cancer Control Programme (CRCCP) and the National Breast and Cervical Cancer Early Detection Programme (NBCCEDP).[15] However, significant proportions of individuals eligible for these programmes do not participate (eg, through fear or not prioritising time to attend for screening),[16] and comprehensive screening programmes are costly to implement, especially in resource-poor settings or low-income and middle-income countries. Other approaches include public health campaigns to encourage individuals experiencing particular symptoms such as weight loss, anorexia and fatigue to visit their family doctors.[17] However, patient help-seeking around cancer is complex, multistaged, and often leads to long delays of weeks or even months.[18] Patients find it hard to interpret and recognise symptoms, with fears of embarrassment and having a potentially fatal or painful condition contributing to long and avoidable delays in help-seeking from health professionals.[18 19] Patients often do not seek help from health professionals for early cancer symptoms, notably from general family physicians, for many reasons including a complex mix of fear, worry and of 'wasting' health professionals' time[19] or due to the high costs of medical care, a lack of health insurance or time constraints.[20]

### Detecting future risk of cancer by modelling data

Screening approaches represent a patient identification (or 'phenotyping' problem) that aims to detect whether the individual has cancer at a particular point in time. However, the ultimate goal of cancer prediction is to determine whether an individual will develop cancer at some point in the future. A simple approach is to stratify populations according to the presence and absence of risk factors, which have been extensively characterised for most cancer types through epidemiological studies over many decades. For example, age, gender, ethnicity, family history and lifestyle factors are well-established risk factors for many types of cancer.[21] The cancer prediction problem can either be regarded as a regression problem, where the input variables are clinical-demographic variables and the output variable is the probability of developing cancer at some point in the future, or as a binary classification problem to determine whether or not a patient will develop cancer at a specific point in time.

### Big data and machine learning for medical prediction models

Advances in digital medicine and computational science have altered the landscape of data available for cancer risk prediction models. For example, in the data-driven healthcare era, there is an increasing amount of 'big' medical data, as most individuals have had interactions with the healthcare system where data is collected in the form of electronic health records (EHRs), which are systematic collections of longitudinal patient health data collected in real time.[22 23] Such large datasets provide powerful new opportunities to develop and refine predictive models and to explore potentially unknown predictor variables.[22] Leveraging often massive amounts of data generated from large populations, much of which may be unstructured, and building optimal models requires the exploitation of advanced computational tools and supporting infrastructure. Machine learning (ML) is a branch of artificial intelligence and an extension of traditional statistical techniques that uses computational resources to detect underlying patterns in high-dimensional data, and it is increasingly being used in different areas of medicine requiring predictions.[24] For example, ML has successfully been used with EHR data to predict incident hypertension[25] and incident chronic kidney disease,[26] and wider popular uses of ML in medicine include the automatic interpretation of medical images such as in radiology[27] and histopathology[28] images.

### A brief description of ML

A comprehensive description of ML models is beyond the scope of this scoping review. However, relevant ML techniques relate to the problem of learning from data samples (eg, EHR data) rather than being preprogrammed with existing knowledge or rules. ML models can either be supervised (ie, where the data are labelled and the algorithm uses these data to learn to predict the output) or unsupervised (ie, where the data are unlabelled and the algorithm learns a structure inherent in the data).[29] The cancer prediction problem is, therefore, a supervised problem; examples are provided as inputs (or features) such as cancer risk factors like age, history, ethnicity or

blood count parameters and outputs (or labels) such as whether or not the individual subsequently develops cancer. A variety of available algorithms learn the best way to map the features to the labels by learning from the observations.[30 31] The resulting model, ideally, will then be able to generalise the information so that it can be applied with high precision to new and unseen data.[30 31]

Some of the main supervised ML models used in medical applications include decision trees (DTs; and their adaptation, random forests (RFs)), support vector machines (SVMs) and artificial neural networks (ANNs).[30 31] DTs produce an output similar to a flow chart formed from feature nodes (risk variables) that best discriminate between different labels (future cancer occurrence) to split the tree.[30 31] In this way, new cases can be assessed by traversing the tree based on the feature values to determine the output for that example.[30 31] DTs are easy to interpret, since users are usually able to visualise the steps leading to a particular classification, which may be useful in a clinical setting where experts might wish to see how a particular decision was made.[30 31] In RFs, several trees are built using subsets of data and features, with predictions decided based on majority voting after the example is assessed with respect to all the constructed trees.[30 31]

In SVMs, each feature (risk factor) is mapped into a higher-dimensional space and the hyperplane that optimally separates the output (future cancer occurrence) modelled.[30 31] SVMs tend to generalise well to unseen data and work well with complex (multidimensional) data but can be hard to interpret.[30 31]

ANNs are inspired by the neural connections in the human brain and are developed by creating nodes (neurons) that weight certain features and produce an output value.[30 31] By layering nodes in between the input layer (features; cancer risk factors) and output layer (label; future cancer occurrence) and modifying the weights during learning through a process called back-propagation, the resulting model forms a prediction for unseen data when one of the nodes in the output layer is positive.[30 31] The terms 'deep neural network' and 'deep learning' are applied to ANNs with large numbers of layers.[30 31] While proving extremely powerful across a range of applications, ANNs can be computationally very expensive and the way in which they classify (ie, the intermediate 'hidden' layers) is opaque, making it difficult to determine exactly how they performed the classification problem.[30 31]

### Rationale for performing a scoping review

ML remains a relatively recent field, so it is unclear exactly to what extent advances have impacted specific healthcare domains. There are currently no extended systematic reviews or scoping reviews on the application of ML to cancer risk prediction in asymptomatic individuals. This prompted us to perform a scoping review of studies using supervised ML techniques to predict the future risk of developing cancer or specific cancers within a general asymptomatic adult (≥18 years) population using clinical and/or demographic and/or basic laboratory data (eg, complete blood counts (CBC)) that are likely to be readily available within the primary care setting. This approach, therefore, allowed to: (1) identify the types of evidence available; (2) clarify key concepts and definitions; (3) examine how research is currently being conducted and (4) to identify knowledge gaps.[32]

### Objectives

The objective of this study was to perform a scoping review and to synthesise knowledge of the nature and effects of current ML techniques for early cancer detection in asymptomatic adults. The scoping review was guided by the following research questions:

1. Which, if any, ML methods are being developed for cancer risk prediction in asymptomatic individuals in the community?
2. How do these models perform compare to each other?
3. Which research or knowledge gaps need to be addressed in order to advance the field?

## METHODS

### Inclusion and exclusion criteria

We used the population, concept and context approach[33] with the following inclusion criteria: (1) general adult (≥18 years) population, either sex, asymptomatic (population); (2) any study using ML techniques to derive predictive models for future cancer risk using clinical and/or demographic and/or basic laboratory data carried out prior to 7 August 2020 (concept) and (3) original research articles conducted in all settings in any region of the world (context).

For the purposes of this study, and recognising that 'ML' algorithms fall along a continuum with statistical techniques,[34] all modelling approaches were included were defined as ML in the respective papers (such as logistic regression (LR)). Exclusion criteria were any ML model used to predict future events in patients with pre-existing or symptoms of cancer; ML models developed using specialised tests such as genetic profiling or imaging tests not generally available in the community; unsupervised ML models; and studies not written in English.

### Literature search

To identify relevant studies, the PubMed database was searched from inception through to November 10, 2020 using the search string: ("Cancer" Or "Cancers" OR "Oncology") AND ("Machine Learning" OR "ML" OR "Data Mining" OR "Decision Support System" OR "Clinical Support System" OR "Classification" OR "Regression" OR "Support vector machines" OR "Gaussian process" OR "Neural networks" OR "Logical learning" OR "Bayesian network" OR "linear model") AND ("prognosis" OR "prognostic estimate" OR "predictor" OR "prediction" OR "model" OR "diagnosis" OR "diagnostic"). This search was supplemented with manual searching of the references and citations of previously published studies. All abstracts identified by the initial search were screened

for inclusion and checked for accuracy. For the included studies, data were extracted from full papers. In instances where more information was required to determine inclusion, the full text of the article was retrieved and assessed against the eligibility criteria.

## Study assessment

The quality of the included studies was assessed using the Newcastle Ottawa Scale (NOS) for observational studies included in the review.[35] The strength of the predictive ability of the included models was assessed using area under the receiver operating characteristics curve (AUC) data, a valid measure for evaluating classification algorithms and one that has been used to compare different algorithms in other meta-analyses.[36 37]

## Patient and public involvement

This study was not explicitly informed by patient priorities, experiences and preferences, although the application of predictive models to assess cancer risk would have a direct bearing on identifying those most at risk and implementing investigations in a timely manner. No patients were involved in the design or conduct of the study and since this was a scoping review of the literature, there were no study participants.

## RESULTS
### Main findings
#### Identified risk models

Using the search strategy, 627 initial studies were identified where 10 studies met the inclusion criteria (table 1 and figure 1).[31 38–47] The most common reasons for exclusion of studies were: (1) models were derived to predict outcomes or responses to therapy in patients with pre-existing cancer and/or (2) the studies used features other than clinical and/or demographic and/or basic laboratory data, such as genetic biomarkers. All studies were retrospective cohort or case-control studies conducted between 2011 and 2020, with 8 out of 10 studies completed in the last 2 years. Eight studies were conducted in the USA and two in Taiwan. One model was built for breast cancer, three for colorectal cancer, one for lung cancer, one for melanoma, two for non-melanoma skin cancer, one for pancreatic cancer and one a general cancer prediction model. Two studies performed external validations of a previously developed colorectal cancer prediction model (table 1).[39 43] In terms of quality assessment, four studies were graded as 'good' quality by the NOS,[39 43 44 46] while six studies were graded as 'poor', in all cases due to comparability of cohorts on the basis of the design or analysis adequately controlling for confounders.[31 38 40 42 45 47]

## Development of the risk models

The models developed in the studies employed a wide range of ML techniques. Two studies compared different modelling approaches on the same dataset,[41 44] while the other eight developed a model using a single approach. The following ML approaches were used: ANNs (8 out of 10 studies), LR (2/10 studies), Gaussian naïve Bayes (1 out of 10 studies), Bayesian network inference (1/10 studies), DTs (1/10 studies) and RFs (2/10 studies), linear discriminant analysis (LDA) (1/10 studies), and SVMs (1/10 studies) (table 1). Data for training and testing were from medical insurance databases (4/10 studies), EHR data repositories (3/10 studies), surveys (2/10 studies) or represented a retrospective analysis of prospectively collected data from a clinical trial (1/10 studies).

As a result of the diverse cancer types being modelled, study aims and the available data, a range of different predictors, features and/or risk factors were included the developed predictive models, which can be grouped into the following categories: (1) patient demographic data, for example, age, gender, ethnicity, family history; (2) social and lifestyle data, for example, cigarette smoking and intensity of exercise; (3) comorbidities, for example, diabetes mellitus, hypertension, congestive heart failure and chronic obstructive pulmonary disease; (4) clinical and practice data, for example, World Health Organisation - Anatomical Therapeutic Chemical (WHO-ATC) prescription codes and clinical encounters and (5) laboratory tests, for example, CBC (table 1). The models that automatically extracted features from EHR records used features that were not always explicitly defined in the respective articles.

## Discrimination and calibration of the risk models

All studies provided AUC values as an assessment of model performance. Calibration (ie, whether the risk estimates were accurate), was not assessed in any study. Two models with particularly high AUC values were the Bayesian network inference model developed by Zhao et al[47], which used 20 demographic, lifestyle, symptom, comorbidity and lab test results to predict the risk of pancreatic cancer with an AUC of 0.91, and the CRC predictive model developed by Wang et al[46], which used a CNN learning on 1929 features (1099 International Classification of Diseases, Ninth Revision (ICD-9) codes and 830 ATC codes). Models with particularly low AUC values were the range of models (LR, Gaussian naive Bayes, DT, LDA, SVM and feed-forward ANN) developed by Stark et al[44]; however, as discussed below, although these models only had AUCs between 0.51 and 0.61, two of the models compared favourably with the BRCAT clinical risk tool.

## Comparison of the risk models with existing predictive algorithms

Hundreds of risk prediction models have been published in the literature for every cancer type, and some of these are already used in clinical practice. It is, therefore important to understand whether the performance of the newer ML-based cancer risk models is comparable to that of existing predictive algorithms. We, therefore, specifically examined whether the studies compared their ML

**Table 1** Summary of studies investigating ML approaches for early cancer detection

| Type of cancer | Reference | Year | Country | Method | Sample | Input | Validation | Performance | NOS | Notes |
|---|---|---|---|---|---|---|---|---|---|---|
| Breast | Stark et al[44] | 2019 | USA | LR, Gaussian NB, DT, LDA, SVM and feed-forward ANN | 1343 breast cancer and 63396 non-breast cancer cases (PLCO Cancer Screening Trial dataset) | Age, age at menarche, age at first live birth, no of first-degree relatives who have had breast cancer, ethnicity, age at menopause, an indicator of current hormone usage, number of years of hormone usage, BMI, pack years of cigarettes smoked, years of birth control usage, number of live births, and an indicator of personal prior history of cancer | 20% testing data (269 breast cancer and 12679 non-breast cancer cases) | LR 0.61 (0.58–0.65); NB 0.59 (0.56–0.62); DT 0.51 (0.50–0.52); LDA 0.61 (0.58–0.65); SVM 0.52 (0.48–0.55); NN 0.61 (0.57–0.64) | 9 (Good) | At an 0.05 level, the LR, LDA, and NN with the broader set of inputs were all significantly stronger than the BCRAT |
| Colorectal cancer | Hornbrook et al[39] | 2017 | USA | ColonFlag ML model | 17095 US community-based insured adults (16195 controls, 900 cases) (insurance data) | Age, gender and blood count panel parameters | Study was a validation of a previously derived model[61] | AUC 0.80 (0.79–0.82) | 7 (Good) | |
| Colorectal | Wang et al[46] | 2019 | Taiwan | CNN | 10185 with CRC, 47967 controls (insurance data) | ICD-9 diagnostic codes, WHO-ATC prescription codes | 5-fold cross-validation | AUC 0.92 | 7 (Good) | |
| Colorectal cancer | Schneider et al[43] | 2020 | USA | ColonFlag ML model | 308721 insurance health plan members (insurance data) | Age, gender and blood count panel parameters | Study was a validation of a previously derived model[61] | AUC 0.78 (95% CI 0.77 to 0.78) | 8 (Good) | The algorithm's accuracy decreased with the time interval between blood test result and CRC diagnosis |
| General | Miotto et al[40] | 2016 | USA | Deep NN and RFs | Model training on 704 587, testing on 76214 (EHR data) | Features extracted from EHR records | Testing on 76214 | Colorectal cancer AUC 0.89, liver cancer 0.89, prostate cancer 0.86 | 6 (Poor) | Outperformed RawFeat and PCA |
| Lung | Hart et al[38] | 2018 | USA | ANN | 1997–2015 National Health Interview Survey adult data; 648 cancer and 488418 non-cancer cases (survey data) | Gender, age, BMI, diabetes, smoking status, emphysema, asthma, ethnicity, Hispanic ethnicity, hypertension, heart diseases, vigorous exercise habits and history of stroke | 30% of data; 195 lung cancer cases and 146524 never cancer cases | AUC 0.86 (training; 95% CI 0.85 to 0.88) and 0.86 (validation; 95% CI 0.84 to 0.89) | 6 (Poor) | RFs and SVM also applied which trained well (RF AUC of 1.00 (95% CI 1.00 to 1.00) and SVM AUC of 0.96 (95% CI 0.95 to 0.97). However, not generalisable: AUC SVM 0.55 (95% CI 0.51 to 0.58); AUC RF 0.81 (95% CI 0.78 to 0.84). |
| Melanoma | Richter and Khoshgoftaar[41] | 2019 | USA | LR, RF, XGBoost | 4 061 172 patients, 10129 with melanoma (EHR data) | Features extracted from EHR records | Fivefold cross-validation | AUC LR 0.76; AUC RF 0.69; AUC XGBoost 0.80 | 7 (Poor) | Smaller amounts of data improved the AUCs |

Continued

**Table 1** Continued

| Type of cancer | Reference | Year | Country | Method | Sample | Input | Validation | Performance | NOS | Notes |
|---|---|---|---|---|---|---|---|---|---|---|
| Non-melanoma skin cancer | Roffman et al[42] | 2018 | USA | ANN | 1997–2015 NHIS adult survey data, 2056 NMSC and 460 574 non-cancer cases (survey data) | Gender, age, BMI, diabetes, smoking status, emphysema, asthma, ethnicity, Hispanic ethnicity, hypertension, heart diseases, vigorous exercise habits and history of stroke | 30% for validation (752 NMSC cases and 138 172 never cancer cases) | AUC values of 0.81 (training, 95% CI 0.80 to 0.82) and 0.81 (validation, 95% CI 0.79 to 0.82) | 6 (Poor) | |
| Non-melanoma skin cancer | Wang et al[45] | 2019 | Taiwan | CNN | 1829 patients with nonmelanoma skin cancer as their first diagnosed cancer and 7665 random controls (insurance data) | Age, sex, ICD-9 diagnostic codes, WHO-ATC prescription codes, and the total numbers of clinical encounters | Fivefold cross-validation | AUC 0.89 (0.87–0.91) | 6 (Poor) | |
| Pancreatic | Zhao et al[47] | 2011 | USA | Bayesian network inference | 98 cases and 14 971 controls (EHR data) | Demographics, lifestyle, symptoms, comorbidities and lab test results (20 variables) | Null | 0.91 (0.87–0.95) | 4 (Poor) | |

ANN, artificial neural network; AUC, area under the curve; BCRAT, Breast Cancer Risk Prediction Tool; BMI, body mass index; CNN, convolutional neural network; CRC, colorectal cancer; DT, decision tree; EHR, electronic health record; ICD-9, International Classification of Disease, Ninth Revision; LDA, linear discriminant analysis; LR, logistic regression; ML, machine learning; NB, naive bayes; NMSC, non-melanoma skin cancer; NN, neural network; NOS, Newcastle Ottawa scale; PCA, principal component analysis; PLCO, Prostate, Lung, Colorectal and Ovarian ; RF, random forest; SVM, support vector machine; WHO-ATC, World Health Organisation - Anatomical Therapeutic Chemical.

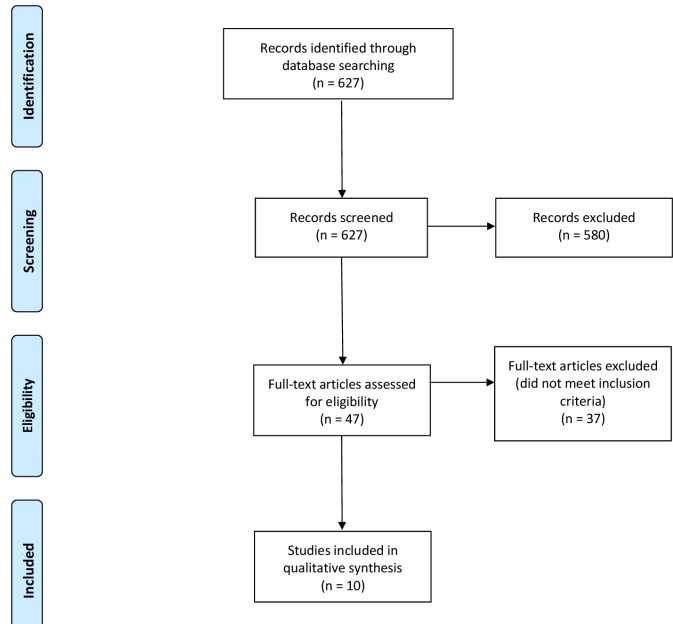

**Figure 1** PRISMA flow chart depicting the search strategy. PRISMA, Preferred Reporting Items for Systematic Reviews and Meta-Analyses.

algorithms with existing algorithms or, if not, how model performance as described by AUCs compared with other published data, despite the limitations of this approach (see below).

Stark et al[44] compared their ML models with an existing clinical prediction tool, the Breast Cancer Risk Prediction Tool (BCRAT; https://bcrisktool.cancer.gov/). The BCRAT tool is an implementation of the Gail model,[48] which is a statistical model that estimates 5-year breast cancer risk in women without a personal history of breast cancer and without known mutations in high-risk breast cancer genes such as *BRCA1* and *BRCA2*. In the Gail model, patients self-report their current age, age at menarche, age at first live birth, number of first-degree relatives who have had breast cancer, ethnicity, and number of previous breast biopsies, variables which are weighted within the model by LR.[48] In addition, BCRAT uses data on a personal history of atypical hyperplasia, where available. Although the AUC values for the models (LR, naïve Bayes, DTs, LDA, SVM and an ANN) tested using a broader set of features than BCRAT were only between 0.51 (DT) and 0.61 (LR, LDA, and ANN), four of the six models (LR, NB, LDA and ANN) outperformed BCRAT (AUC 0.56). Other metrics were also used to assess model performance (sensitivity, specificity and precision), which were comparable between the ML algorithms and the BCRAT, and both BCRAT and the ML models had low precision (~2%). Furthermore, when comparing the different ML models, LR and LDA produced higher AUCs than the ANN model, despite the potential for ANNs to better model noisy data and complex non-linear functions.[49] The authors suggested that this might have been due to the limited amount of

available training data or the selection of hyperparameters.[44] It was observed that (1) the derived ML models using an extended and set of features available in primary care can deliver improvements on current clinical algorithms; (2) that adding additional features has a greater impact on improving model performance (ie, higher AUC) rather than simply using more complex models and (3) that AUC values must be interpreted in the context of existing methods, such as existing, clinically used risk prediction models such as the BCRAT or Gail model, rather than in isolation.

In a systematic review of 52 colorectal cancer models predicting future risk of disease in asymptomatic individuals,[50] 37 models reported AUC values, which ranged from 0.65 and 0.75. These included five models that used routine data exclusively and did not include questionnaires or genetic biomarkers. In comparison, the AUC values for ColonFlag,[39 43] an ML model that uses age, gender and CBC features to predict the future occurrence of colorectal cancer up to 12 months prior to diagnosis, were 0.78–0.82.

In another systematic review involving 25 risk prediction models for lung cancer that used only epidemiological parameters as input (ie, no laboratory parameters),[51] AUCs ranged between 0.57 and 0.86, which compares to an AUC of 0.86 (in both training and validation cohorts) for the ANN model developed by Hart et al.[38] In their systematic review of 25 melanoma risk prediction models, Usher-Smith et al[52] showed in a summary ROC curve that most models had similar discrimination of 0.76, which compares to the highest AUC of 0.80 achieved using XGBoost ML by Richter et al[41].

## DISCUSSION
### Strengths and limitations of existing ML approaches
The reviewed studies reviewed highlight that several different techniques have successfully been used to develop models and that ML can be applied to large-scale insurance and EHR data containing hundreds or thousands of features in order to build predictive models. However, the survey also highlights a number of gaps in the application of ML to predicting the risk of future cancer in asymptomatic individuals. These can be divided into those relating to: (1) study populations; (2) model types and comparisons and (3) model validation, comparisons and calibration.

### Study populations
To date, ML techniques have only been applied to or validated in datasets from developed countries, representing a fraction of the overall global population and their dietary and lifestyle factors. Given that the aetiology of cancer, risk factors and genetics differ in different populations,[53] models developed in populations in high-income countries may not be generalisable to those from low- and middle-income countries (LMICs). The development and validation of models in LMICs could have two advantages:

first, it would determine the generalisability (and therefore utility) of that model in other populations, better serving the needs of individuals in LMICs; second, disparities between models developed in different geographical settings could provide valuable new information about factors contributing to cancer risk. Generalising risk prediction models is likely to be challenging, since resource-poor countries often do not have the necessary infrastructure nor the epidemiological research capabilities of institutions in high-income countries.

Furthermore, current ML models predict the risk of a limited number of cancer types. Although breast, colorectal and lung cancer are the three most common cancers and therefore account for a large proportion of overall cancer burden, it is still important to detect all cancers early. This is especially true for those cancers that are usually silent (asymptomatic) for long periods of time, present late with advanced-stage disease, and for which there are currently no screening programmes in place, such as ovarian and pancreatic cancer. Predicting future risk of these cancers could allow closer monitoring of at-risk individuals.

## Model types and comparisons

A wide variety of ML methodologies have been applied and, despite being applied to the same research problem, this scoping review has not identified a single 'best' method. Two issues arose in studies that compared different ML approaches on the same datasets. First, although different models had similar AUCs during training, not all models generalised well to validation datasets; robust model validation is therefore important to ensure model validity (see below). Second, although in general it is assumed that larger amounts of training data improve model performance,[54] Richter and Khoshgoftaar[41] found that equivalent or even better model performance was achievable using reduced datasets (hundreds of thousands vs millions of datapoints). This might be due to high levels of homogeneity in the 'no cancer' class, resulting in fewer instances being required to produce a generalisable model, or as a result of overfitting. Although the requirement for less data for the cancer prediction problem could make ML techniques more accessible to researchers without extensive computing infrastructure and allow smaller datasets to be leveraged for model construction, ML requires over 10 times the amount of data per variable for stable discrimination compared with traditional approaches such as LR.[54] Instead of regarding data requirements as 'too high' or 'too low', it might be better to consider how much data is required for a particular predictive context. Riley et al[55] recently provided an implementation of how to calculate the sample size required to develop specific clinical prediction models, which will help researchers prospectively plan their in silico experiments and avoid using datasets that are too small for the total number of participants or outcome events.

## Model validation, comparisons and performance evaluation

With the exception of the two studies evaluating a previously defined algorithm for colorectal cancer, no other study used external validation datasets to assess model generalisability, instead opting for either a single holdout validation sample or fivefold cross-validation. While useful for assessing overfitting,[56] these approaches do not account for population bias in the training dataset nor differences in other target populations. Studies seeking to develop ML models should seek to validate models in independent populations, recognising that an advantage of an 'ungeneralisable' model might be insights into cancer risk in other populations. Furthermore, since physicians may code diseases in EHRs differently over time (for instance, due to altered management or incentives), even initially generalisable models may need revalidation over time.[23 57]

Discrimination (ie, the ability to distinguish a patient with a high(er) risk of developing cancer from one with a low(er) risk of developing cancer) was measured in every study using the AUC, as is common in the field. However, discrimination is not the only metric of model performance.[58] Another important measure of model performance, particularly for the clinical setting, is calibration, that is, establishing that the risk estimates are accurate.[59] In this setting, this means that the model should not unduly overestimate or underestimate the risk that a patient will develop cancer; to do so would mean that a patient might be subjected to investigations and the associated worry of their likelihood of developing cancer (overestimated risk), or, conversely, underinvestigated and falsely reassured in the case of underestimated risk. Therefore, a highly discriminatory but poorly calibrated model is likely to have poor clinical utility.

None of the studies reviewed here performed calibration analysis, which is not uncommon in this field. Indeed, in their systematic review of 71 studies using ML for clinical prediction for a wide variety of clinical purposes, Christodoulou et al reported that 79% of studies failed to address the calibration problem.[37] Therefore, caution must be applied when interpreting and comparing the performance of current ML models based on AUC alone, since is an incomplete measure of performance that must be considered together with methodological aspects such overfitting, measurement error, and population heterogeneity that might influence the estimation of predictive performance.[37 59]

## Implications for clinical practice

The ML models described in this scoping review generally show high AUC values. So, are any of these models ready for clinical use? The ColonFlag model[39 43] is an example has recently been implemented at Barts Health NHS Trust[60] to identify patients at particularly high risk of CRC, particularly as clinicians struggle to prioritise patients in the backlog created by the Coronaviruses (COVID-19) pandemic. The ColonFlag model is the

only model identified in this scoping review that has undergone extensive external validation in independent datasets.

New ML models need to be contextualised with currently available best clinical practice in order to fully evaluate their potential clinical value. Comparing the relatively poor AUC values of the Stark *et al*[44] models with BCRAT revealed that they in fact outperformed it in many cases. In their comparison of their ANN with screening methods for lung cancer such as low-dose CT, chest X-ray and sputum cytology, Hart *et al*[38] noted that (according to sensitivity and specificity) it outperformed most of the other available non-invasive methods. Thorough side-by-side comparisons of newly developed models with other prediction tools would be helpful in establishing future clinical utility.

Finally, this scoping review highlights that model performance should not be evaluated solely on the basis of AUC values but also in terms of other importance performance metrics such as calibration, without which a model might inaccurately assess risk and therefore prompt inappropriate management.

### Unanswered questions and future research

The few models that are currently available are methodologically diverse, rarely validated in independent datasets to ensure generalisability, and do not cover all cancer types. Even if ML techniques offer only small improvements in cancer detection rates, these improvements are likely to be of high clinical significance given the large size of the global population with or at high risk of cancer and the high mortality and costs associated with late cancer diagnoses.

However, the scoping review identifies a number of research gaps that will need to be addressed in order to deliver validated ML-based models to assist clinical decision making. First, future studies must take steps to establish model generalisability through validation in independent cohorts, including those from LMICs. Although the latter may be challenging, it could be argued that even negative generalisability studies might provide an opportunity to learn more about cancer risk factors in different populations. Second, the scoping review fails to establish which ML approach best suits the cancer prediction problem but does show that, where possible, side-by-side comparisons of different methods can reveal important information about generalisability as well as performance and that these comparisons are desirable whenever possible. Third, many important cancer types, particularly 'silent killers' like ovarian cancer, have currently not been the subject of ML modelling approaches; ML could provide an important, low-cost, non-invasive method to identify individuals at high risk of clinically silent cancers that require closer monitoring. Fourth, progress has been made in defining approaches to tailor sample sizes to the specific setting of interest to minimise overfitting and targeting precise estimates of key parameters, and these principles must be applied when testing and validating models to ensure robust model performance. Finally, ML models need to be compared with the best available clinical tools so that their potential clinical utility is transparent.

### Limitations of this study

Our study has a number of limitations. First, despite recognising the need for a scoping review due to the paucity of literature on the topic, we were only able to identify ten papers meeting the inclusion criteria. It is therefore difficult to draw definitive conclusions about the performance of these models. Furthermore, although AUC values provide an indication of how discriminative the models are, they do not allow for meaningful comparisons of models trained and evaluated on different datasets. Six out of ten studies were defined as poor quality due to a lack of controlling for confounders in the study design, which may have introduced significant bias. Finally, we only search the PubMed database and articles published in English, so some papers in other languages or in databases for non-medical disciplines may have been missed.

### CONCLUSIONS

This scoping review highlights that applying ML to cancer prediction is a promising field provided that the identified issues such as generalisability, validation and clinical applicability, model calibration and dataset selection are addressed in future studies. We hope that the identified research gaps focus future research efforts to deliver validated ML-based models to assist and improve clinical decision making

**Contributors** AAA defined the research question of the scoping review, conducted the literature search and summarised the findings. HK and AGL supervised the research. All authors drafted and revised the manuscript.

**Funding** AGL is supported by funding from the Wellcome Trust (204841/Z/16/Z), National Institute for Health Research (NIHR) University College London Hospitals Biomedical Research Centre (BRC714/HI/RW/101440), NIHR Great Ormond Street Hospital Biomedical Research Centre (19RX02).

**Competing interests** None declared.

**Patient and public involvement** Patients and/or the public were not involved in the design, or conduct, or reporting, or dissemination plans of this research.

**Patient consent for publication** Not required.

**Ethics approval** Not applicable. This study does not involve human participants nor animal subjects.

**Provenance and peer review** Not commissioned; externally peer reviewed.

**Data availability statement** Data are available in a public, open access repository. Only public published papers were used. No confidential data.

**ORCID iDs**
Asma Abdullah Alfayez http://orcid.org/0000-0002-7448-2095
Holger Kunz http://orcid.org/0000-0002-2229-4500
Alvina Grace Lai http://orcid.org/0000-0001-8960-8095

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
