## [Reviewer comments · BMJ Open]

ARTICLE DETAILS

TITLE (PROVISIONAL)	Predicting the risk of cancer in adults using supervised machine learning: a scoping review
AUTHORS	Alfayez, Asma; Kunz, Holger; Lai, Alvina

VERSION 1 – REVIEW

REVIEWER	Nindrea, Ricvan Universitas Andalas
REVIEW RETURNED	27-Jan-2021

GENERAL COMMENTS	1. More details should provided for novelty of research2. Add details of quality assessment for articles included3. Add more limitation of study4. Add more suggestion for future research
---

REVIEWER	Christodoulou, Evangelia DKFZ, Cancer Epidemiology
REVIEW RETURNED	22-Feb-2021

GENERAL COMMENTS	This is an interesting scoping review, which besides the limited number of included studies, brings out the important message that particular attention to the external validation of clinical prediction models using ML algorithms is necessary. Major comments 1. The assessment of the predictive performance of a model only in terms of AUC is not enough. Please derive information on calibration from the included studies as well.2. I would not agree with the categorisation of AUC values into “excellent”, “good”, “fair” and so on. This forces arbitrariness in assessing the predictive performance of the models. At the same time, an observed AUC of 1 should be alarming in terms of the methodological aspects of model development (most likely overfitting is present). Moreover, it is dangerous to characterize model performance only in terms of the presented AUC values without delving into the potential methodological flaws regarding model development and validation, that can be over- or underestimating the predictive performance. Lastly, the included studies are too few to even create the need for such a categorisation of AUC values.
---

	3. Table 1: It is not clearly stated how for studies that are using multiple ML algorithms, a general characterization of model performance as “poor” or “excellent” has been derived. Please either remove the column “Model performance” or be explicit about which model each characterization corresponds to. 4. Line 290-291: I do not think that this scoping review has gathered enough evidence to make such claims, given the absence of qualitative assessment of the included studies and the fact that only 10 have met the inclusion criteria. 5. Lines 325-327: I do not see how the requirement for less data for cancer prediction could make ML techniques more valuable for researchers. This contradicts reference 54 in your manuscript, where ML methods have been observed to require a larger amount of data in order to perform satisfactorily. Perhaps your point here was that the high extent of homogeneity of data used in cancer prediction alongside with the presence of well-known strong predictors, can allow a model to reach satisfactory performance levels with the use of less data compared to other clinical prediction problems. 6. Lines 371-373: Again, please reword this. The amount of data that a prediction model needs to reach a “decent” performance is dependent on a wide variety of reasons (please check Riley et al BMJ 2020). 7. Line 380: Please remove this claim. This scoping review includes too few studies to form such inference and at the same time, the included models have not undergone qualitative assessment. Minor comments 8. Line 173: Please try to reword the first sentence and not start a new paragraph with “Therefore”. 9. Line 207: Please reword this line, because I find “predict prognosis” to be confusing. Perhaps you can phrase it as such “models were derived to predict final outcomes or responses...”
--	--

REVIEWER	Stark, Gigi F. Yale University
REVIEW RETURNED	30-Jun-2021

GENERAL COMMENTS	Overall - Overall, the paper effectively identifies both existing ML cancer prediction approaches and potential gaps in research. Moreover, it is well written.
---

- The paper's comparison of performance of ML models to each other could be improved in two ways: 1) removing the excellent, good, fair, poor, and fail categories for AUC as AUC depends heavily on confounding factors from the data set and thus, broad labels oversimplify comparisons between the AUCs of models trained and evaluated on different data sets, and 2) more explicitly articulating and supporting the goal of the "Comparison of the risk models with existing predictive algorithms" section. I have provided further comments on these points and other minor suggestions below.

Introduction

- [Pages 3-4] To improve the beginning of the Introduction section, the author should consider moving the page 4 paragraph on cost saving for early-stage cancer (the paragraph that begins with "Globally, treatment for early-stage cancer confer significant cost-saving benefits") to the end of the page 3 paragraph that begins with "Early cancer diagnosis is associated with significantly higher survival rate and lower mortality and associated costs" and also discusses the economic burdens of cancer treatment.

- [Pages 5-6] On page 5, the author states, "The cancer prediction problem can either be regarded as a supervised learning problem where the input variables are clinical-demographic variables and the output variable is the probability of developing cancer at some point in the future or as a binary classification problem to determine whether or not a patient will develop cancer at a specific point in time." However, on page 6, the author appropriately states, "The cancer prediction problem is therefore a supervised problem." On page 5, it seems that rather than stating that cancer prediction can be either a "supervised learning" or a "binary classification" problem, the author should state that cancer prediction can be either a "regression" or a "binary classification" problem.

Objectives

- [Page 7] This section is clear, organized, and to the point and thus, sets up the rest of the paper well!

Methods

- [Pages 8-9] As detailed in the above "Overall" section, absolute AUC can depend on a variety of confounding factors related to the data set (e.g., data set size, percentage of positive vs. negative cases in the data set, etc.). Thus, direct comparisons of the AUCs of models trained and evaluated on different data sets are not necessarily meaningful. While highlighting the models that are particularly strong is valuable, creating excellent, good, fair, poor, and fail categories for AUC ranges is not appropriate, and these categories should be removed.

Results

	- [Pages 16-17] The goal of the "Comparison of the risk models with existing predictive algorithms" section is unclear. The author should better articulate the key message in this section and link the message more clearly to the cited content. Discussion - [Page 18] The "Study populations" section touches on a lot of critical and interesting topics. - [Page 19]: I concur with the author's point: "New ML models need to be contextualised with currently available best clinical practice in order to fully evaluate their potential clinical value." However, categorizing the diverse reviewed models into excellent, good, fair, poor, and fail categories based on AUC does not reflect this point.
--	---

REVIEWER	Clark, Alex University of Alberta, Faculty of Nursing
REVIEW RETURNED	06-Jul-2021

GENERAL COMMENTS	A well-designed systematic review into a topic that is both cutting edge and clinically important. With small improvements, this will be an important and well-cited contribution to an emerging and complex field. Introduction The topic is adequately contextualized and introduced for international readers around cancer. The benefits of early cancer diagnosis are clear and well explained with specific examples for the general reader. The potential contribution of machine learning / modelling is fairly and accurately articulated in a manner unusually clear for technical articles. Some minor editing of wording is needed for clarity and perhaps acronyms could be used less. Scoping review is clearly the right method to answer the research question(s). Abstract This needs to be revised not just to address research gaps- but the actual research questions of the review. Methods The search was comprehensive, well detailed, and mapped directly to the question. Inclusion criteria are clear and appropriate. Findings These are well presented. The review does a good job of synthesising knowledge from the studies – which was widely heterogenous in scope and outcome. Again, acronym use here could be addressed via simple editing. I was not clear on the rationale of including systematic reviews (refs 50, 51) in the cohort of studies – this should be explained more or addressed.
--

Discussion

The findings, strengths, and limitations are well discussed. Clinical considerations are welcome to see. These need to be more integrated with the actual findings of the review.

VERSION 1 – AUTHOR RESPONSE

Reviewer: 1

Dr. Ricvan Nindrea, Universitas Andalas

1. Further justification for performing a scoping review on this topic – and hence the novelty of the research given that this is the first such review – has been added to section “Rationale for performing a scoping review” (lines 152-155).
2. All the articles have now been assessed using the Newcastle Ottawa Scale (lines 197-198), and these results have been added to Table 1 and explained in the results (lines 220-223) and limitations (lines 423-425).
3. We have now added a dedicated limitations section (lines 418-427).
4. We have expanded the future research section, not least due to some of the points raised by other reviewers (lines 412-415).

Reviewer: 2

Dr. Evangelia Christodoulou, DKFZ

Major comments

1. Thank you for raising this important point. We agree that assessing model performance in terms of calibration as well as discrimination is important. We have reviewed the papers, and none of the included studies assessed calibration. We note that this is consistent with your excellent recent paper that showed that 79% of the 71 studies you meta-analysed did not address calibration. We have now reported this result (lines 247-250) and included the issue of calibration as a discussion point in the revised version of the text (lines 358-374) and in the abstract (lines 16-17, 22). Hopefully, together with your own work, this will reinforce the point that to move forward in the field there need to be methodological and reporting standards.
2. Thank you, and the same point was raised by Reviewer 3. We have now removed the AUC categorisations. Furthermore, we have raised the issue of the lack of comparability of AUC values for the reasons you mention in the headline strengths and limitations (lines 34-35) and discussion (lines 390-393).
3. These categorisations have been removed.
4. We agree, and this sentence has been removed.

5. Yes, we agree that this was overstating a point based on a single observation. Therefore, we have modified the discussion to reinforce the more useful point that the data requirements must match the specific clinical problem being solved, with reference to the Riley paper (lines 340-346, lines 412-415).

6. Please see response to point 5.

7. Thank you, we have modified the conclusions accordingly (lines 430-432).

Minor comments

8. Sorry, this was a typo.

9. This clumsy phrasing has been re-worded.

Reviewer: 3

Dr. Gigi F. Stark, Yale University

Overall

- Thank you for these positive comments.
- The paper's comparison of performance of ML models to each other could be improved in two ways: 1) removing the excellent, good, fair, poor, and fail categories for AUC as AUC depends heavily on confounding factors from the data set and thus, broad labels oversimplify comparisons between the AUCs of models trained and evaluated on different data sets, and 2) more explicitly articulating and supporting the goal of the "Comparison of the risk models with existing predictive algorithms" section. I have provided further comments on these points and other minor suggestions below. Please see answers to specific points below.

Introduction

- This has been adjusted as advised.
- Thank you, we agree, and this has been adjusted.

Objectives

- Thank you!

Methods

- Thank you, and the same point was raised by Reviewer 2. We have now removed the AUC categorisations. Furthermore, we have raised the issue of the lack of comparability of AUC values for the reasons you mention in the headline strengths and limitations (lines 34-35) and discussion (lines 390-393).

Results

- We have now written an introductory paragraph which justifies the comparisons (lines 259-264).

Discussion

- Thank you!
- We have now removed the AUC categorisations. Furthermore, we have raised the issue of the lack of comparability of AUC values for the reasons you mention in the headline strengths and limitations (lines 34-35) and discussion (lines 390-393).

Reviewer: 4

Dr. Alex Clark, University of Alberta

Introduction

Thank you. We have only defined acronyms on first use, and ML is used consistently throughout the manuscript.

Abstract

We have subdivided the objectives more clearly for clarity.

Methods

Thank you.

Findings

We have now written an introductory paragraph which justifies the comparisons with existing predictive algorithms (lines 259-264). We compared with meta-analyses since these represent the best of the evidence on existing “traditional” approaches to prediction in multiple cancer types, on which hundreds of papers have been published.

Discussion

The clinical practice section has been expanded in response to other reviewers’ comments and now refers to the specific findings of the review.